# Spontaneous Bacterial Peritonitis: The Incremental Value of a Fast and Direct Bacterial Identification from Ascitic Fluids Inoculated in Blood Culture Bottles by MALDI-TOF MS for a Better Management of Patients

**DOI:** 10.3390/microorganisms10061188

**Published:** 2022-06-09

**Authors:** Romain Lotte, Audrey Courdurié, Alice Gaudart, Audrey Emery, Alicia Chevalier, Albert Tran, Mathilde Payen, Raymond Ruimy

**Affiliations:** 1Laboratory of Bacteriology, Centre Hospitalier Universitaire de Nice, 06200 Nice, France; gaudart.a@chu-nice.fr (A.G.); emery.e@chu-nice.fr (A.E.); chevalier.a3@chu-nice.fr (A.C.); payen.m@chu-nice.fr (M.P.); ruimy.r@chu-nice.fr (R.R.); 2Université Côte d’Azur, Centre Hospitalier Universitaire de Nice, 06000 Nice, France; tran.a@chu-nice.fr; 3Université Côte d’Azur, Inserm, C3M, 06200 Nice, France; 4Infectious Diseases Department, Centre Hospitalier Universitaire de Nice, 06200 Nice, France; courdurie.a@chu-nice.fr; 5Hepatology Department, Centre Hospitalier Universitaire de Nice, 06200 Nice, France

**Keywords:** direct identification of bacteria, MALDI-TOF, ascites

## Abstract

Spontaneous bacterial peritonitis (SBP) is a severe infection that requires fast and accurate antibiotic therapy to improve the patient outcome. Direct bacterial identification using MALDI-TOF mass spectrometry from ascitic fluid inoculated in blood culture bottles (BCBs) could therefore improve patients’ management. We evaluated the impact of the implementation of this method for the treatment of patients. Our identification protocol was performed on 136 positive BCBs collected from 61 patients between December 2018 and December 2020. The therapeutic impact of our protocol was evaluated using a before (2015–2016) and after (2019–2020) case–control study in two populations of 41 patients diagnosed with SBP and treated with antibiotics. The decrease in time to first identification and the optimization of antibiotic therapy following communication of the identification result were evaluated. Our protocol allowed us to identify 78% of bacteria in ascitic fluids. The transmission of the direct identification allowed the introduction or adaption of the antibiotic therapy early in 37% of SBP, with a mean decrease in time to first antibiotic change of 17 h. Our direct identification protocol for positive inoculated ascitic fluids is fast, reliable and inexpensive. Its routine integration into a microbiology laboratory allows the early introduction of appropriate antibiotic therapy and improves the management of patients with SBP.

## 1. Introduction

Spontaneous bacterial peritonitis (SBP) is a common and life-threatening bacterial infection in patients with cirrhosis, associated with significant morbidity and mortality [1,2,3,4]. SBP is defined by the presence of ascitic fluid, absolute neutrophil count >250 cells/mL and the absence of features suggestive of secondary bacterial peritonitis [1]. This infection requires a rapid and efficient antimicrobial treatment. If the perfect empirical antibiotic choice remains controversial, it is now obvious that a rapid bacterial identification to the species level, associated with prompt antibiotic treatment active on the microorganism isolated in culture, could improve patient outcome and reduce mortality [5,6,7]. In a microbiology laboratory, the direct seeding of the original ascitic fluid specimen on an agar plate is routinely performed to identify the bacteria involved in the SBP. However, due to the lack of sensitivity of the direct culture, ascitic fluid specimens are also inoculated in blood culture bottles (BCBs). While increasing the sensitivity of the culture, the time to bacterial identification using matrix-assisted laser desorption ionization time-of-flight mass spectrometry (MALDI-TOF MS) may also increase due to the need to subculture the agar plates before identification. Therefore, direct bacterial identification using MALDI-TOF MS from culture positive liquids inoculated in BCBs could save time in the identification and improve patients’ management.

In this study, we first evaluated the performance of our direct MALDI-TOF identification protocol, previously validated for blood culture samples, for ascitic fluids inoculated in BCBs [8]. Then, for the first time, we determined the decrease in time to first results and estimated the impact of the implementation of this method in the routine workflow of a microbiology laboratory for the antibiotic treatment of the patients with SBP.

## 2. Materials and Methods

### 2.1. Study Design and Samples Procedure

A prospective study was conducted at our 24/7 laboratory of bacteriology at Nice Teaching Hospital (1700-bed tertiary care centre) from December 2018 to December 2020. During this two-year period, a total of 136 ascitic fluids incubated in BCBs (collected in pairs: one aerobic, BacT/ALERT^®^ FA Plus and one anaerobic, BacT/ALERT^®^ FN Plus) respectively sampled from 61 different patients, were detected positive by automated device BacT/ALERT 3D (bioMérieux, Marcy l’Etoile, France). All bottles were incubated for up to 5 days at 37 °C until they were flagged as positive. Every positive BCB was Gram stained using the PREVI^®^ Color automated Gram staining system (bioMérieux, Marcy-l’Étoile, France), and agar plates were inoculated (blood, chocolate, Drigalski, CAP) depending on the results of the Gram staining. In addition, when the Gram-stained smear showed Gram-positive cocci in pairs and chains, an optochin disk was added to the chocolate agar plate to differentiate *Streptococcus pneumoniae* from other streptococci. Bacterial identification was peformed by MALDI-TOF MS directly on BCB positive samples using our 5 min in-house extraction method previously validated for bacterial identification on blood culture samples (called “Day0” identification in the rest of the article) [8]. Firstly, 8 drops of BC broth (approximately 200 µL) were added to a 1 mL solution of Triton™ X-100 (Sigma-Aldrich, Lyon, France) at a concentration of 0.1%. The mix was vortexed for 5 s, then centrifuged at 13,000 rpm for 2 min. The supernatant was discarded, then a further 1 mL of Triton 0.1% was added before a second cycle of vortexing and centrifugation. The supernatant was again removed and the pellet was ready for identification using MALDI-TOF MS. This protocol does not allow yeast direct identification, which requires a longer optimized extraction protocol. Identification was also peformed on colonies (subcultured from BCBs) after 18–24 h of incubation on agar plates (called “Day1”). Ascitic fluids were also collected into sterile tubes in accordance with routine procedure and sent to the laboratory for analysis. Gram staining, white blood cell count (WBC) (cell/mm^3^) and red blood cell count (RBC) (cell/mm^3^) were performed on the fluids. Samples were seeded on agar plates (Drigalski, blood, chocolate, Colistine Aztreonam blood agar Plate (CAP), Oxoid Thermo Fisher Scientific, Dardilly, France) and in Schaedler broth medium and incubated under appropriate atmosphere conditions as recommended by the Société Française de Microbiologie and the European Society of Clinical Microbiology and Infectious Diseases.

### 2.2. MALDI-TOF MS

Target plates were read in a Microflex LT (Bruker, Wissembourg, France) with the MALDI Biotyper 3.1 software and Bruker database 5989. The process of identification was performed as previously described [8].

Direct bacterial identifications (Day0) were considered valid to the species level when they were the same as those obtained by the conventional method with a log (score) ≥1.5 and three-times repeatable on at least one of the two spots. This protocol was extensively validated in our previous study [8]. For polymicrobial samples, direct bacterial identification at Day0 was considered valid to the species level when each organism had a log (score) ≥1.5 and three-times repeatable on at least one of the two spots.

To assess the reliability of our identification protocol (previously validated for blood culture samples) for ascitic fluids, the number of correct identifications at Day0 with an equal or higher score than the threshold were divided by the total number of bacteria. Polymicrobial cultures were analysed separately according to the same criteria.

### 2.3. Patients and Data Collection

For each patient included in the study, age, sex, clinical diagnosis, patient outcome and antibiotic regimen were collected from the patients’ medical files. Time of bacterial growth in BCBs and type of bacteria isolated in culture were also collected from the Laboratory Information System (LIS). Regarding the antibiotic treatment, the following data were extracted from the hospital prescription software: the empirical therapy prescribed and the first change in antibiotic therapy, the time to first change in antibiotic therapy occurring after communication of the first microbiological result. Changes included: antibiotic initiation or, in any case of empirical therapy, addition of a new drug, and switch to a different treatment regimen (if multiple antibiotic treatment were initially prescribed).

### 2.4. Evaluation of Decreasing Antibiotic Optimization Turnaround Time

To evaluate the real impact of our direct identification protocol on the time to first change in antibiotic, we performed a two-years before (2015–2016)-and-after (2019–2020) comparative study in two groups of patients diagnosed with SBP and treated by antibiotics (41 patients). Patients diagnosed with bacterascites were not treated with antibiotics and therefore not included in the comparison (20 patients). Since our laboratory was not open 24/7 during the before period (2015–2016), we performed a comparative analysis including only BCBs that flagged positive from 8.30 a.m. to 6.30 p.m. in the “before” or “after” period. We compared the mean time to first antibiotic change in the two populations. In order to calculate the time to first antibiotic change for each patient provided by our technique, we automatically extracted from our LIS, (i) for the “after” period: the time of identification directly performed on BCBs (corresponding to the delay between the time when the BCB was incubated in the device and the time of direct identification at Day0); (ii) for the “before” period: the time of Gram staining directly from BCBs as well as the time of bacterial identification of colonies for the “before” period (Figure 1). The percentage of antibiotic optimization at Day0 was also compared between these two populations.

### 2.5. Statistical Analyses

Data were analyzed with Prism 7.0 (GraphPad Software) by unpaired Student’s t-test, chi-square test and Fischer test (**** *p* < 0.0001, * *p* < 0.05, ns: non-significant). We used the unpaired Student’s t-test to compare the quantitative variables from the two populations (age and mean time to first change in antibiotics), and the chi-square test and Fischer test to compare the categorical variables (sex ratio and % of antibiotic change).

## 3. Results

### 3.1. Analysis of Direct Bacterial Identification by MALDI-TOF MS for Ascitic Fluid

One hundred and thirty six BCBs inoculated with ascitic fluids in which bacteria were identified by the conventional method on Day1 were included in the study and analysed. A total of 118 (87%) were monomicrobial and 18 (13%) were polymicrobial. A total of 156 isolates belonging to 35 bacterial species were identified by MALDI-TOF MS on Day1 (Table 1). Using our in-house extraction protocol, we were able to correctly identify at Day0 at least one bacterium to the species level in 116/136 of the ascitic fluids (85%). The concordant identifications from ascitic fluids on Day0 of the bacterial species definitively identified on Day1 are listed in the Appendix A. In our study, we were able to successfully identify 78% (121/156) of bacteria to the species level for ascitic fluids: 86% of Gram-negative bacteria and 73% of Gram-positive bacteria. Interestingly, we were able to correctly identify 91% of Enterobacteriaceae and 84% of enterococci, which are the two most frequently encountered species in ascitic fluid infections. Our method failed to identify *Streptococcus mitis* group species 0/5. Of note, our protocol allowed us to identify at Day0 at least one bacterial species in 18/18 (100%) of the polymicrobial samples. Moreover, we identified two out of the two species isolated in culture in 3/18 (17%) of the polymicrobial specimens. The antibiotic course of patients diagnosed with polymicrobial SBP are presented in Appendix A.

### 3.2. Impact of Our Method for the Management of Infected Patients: Before-and-After Case–Control Study

Among the 61 patients for whom a BCB inoculated with ascitic fluid was detected positive by the BacT/ALERT^®^ device, 41 were diagnosed with SBP and 20 of them had asymptomatic bacterascites. The twenty patients diagnosed with bacterascites were not included in the comparative analysis as they did not receive any antibiotic treatment. The comparative analysis between two populations of 41 patients treated for SBP before (2015–2016) and after (2019–2020) the implementation of our direct identification method in the workflow of our laboratory, showed promising results. Interestingly, this comparison yielded a significant decrease in the mean time to first change between antibiotics from 41.3 h to 24.3 h (*p* < 0.0001 (****)). Of note, the percentage change in antibiotic regimen following the communication of the first microbiological result at Day0 was 15% (6/41) and 37% (15/41) for “before” and “after” direct identification implementation in our laboratory, respectively (*p* = 0.02 (*)) (Table 2).

During the “after” period of the study (2019–2020), a total of 15 out of 41 patients benefited from an early adaptation of their antibiotic regimen following the direct bacterial identification at Day0. The communication of the result of the bacterial identification at Day0 allowed either the adaptation of the empirical antibiotic therapy active against the identified bacteria for 60% of these patients (9/15), or the de-escalation of the antibiotic regimen (6/15) (Table 3). For 8 out of 15 patients (53%), enterococci or Enterobacteriaceae were responsible for the SBP. For the seven other patients, the bacteria isolated were, Staphylococcus aureus, Streptococcus sp, Listeria monocytogenes and Bacteroides fragilis for 2/15 (13%), 2/15 (13%), 2/15 (13%) and 1/15 (7%) patients, respectively. Sixty seven percent (10/15) had a favourable short-term evolution at one month after infection, while 33% had complications (5/15). Three patients died in the three weeks following the first episode of SBP (Table 3).

## 4. Discussion

In this study, we validated an in-house method of direct bacterial identification by MALDI-TOF MS on ascitic fluids. Indeed, our fast and cost-effective protocol allows a decrease of 17 h in mean time to first antibiotic change, allows the early introduction of appropriate antibiotic therapy and can improve the management of patients diagnosed with SBP.

MALDI-TOF MS is known to be a reliable technique for identifying bacteria from plate cultures using pathogen protein profiles [13,14]. In the literature, some studies have previously evaluated various protocols used for direct bacterial identification by MALDI-TOF on blood culture samples [8,15,16]. Interestingly, these protocols minimize the rendering-time result with a time saving of 1 to 24 h over conventional methods, depending on the extraction technique [8,15,16], which contribute to reducing morbidity [17,18]. However, data regarding the performance and clinical impact of these identification methods on other body fluids including ascitic fluids are scarce.

The first aim of our study was to evaluate our in-house direct identification protocol previously used on BCBs for blood culture samples on ascitic fluids. Interestingly, using the same threshold (log(score) ≥1.5 and three-times repeatable, which allowed 80% of correct identification at Day0) as for blood culture samples, we were able to correctly identify 78% (121/156) of bacteria to the species level for ascitic fluids. To the best of our knowledge, though several authors have attempted to use direct proteomics methods on articular fluids inoculated in BCBs [19,20], this is the first study to evaluate direct bacterial identification by MALDI-TOF MS on ascitic fluids. Interestingly, we were able to correctly identify 91% of *Enterobacteriaceae* and 84% of enterococci, which are the two most frequently encountered species in ascitic fluid infections. Remarkably, at least one bacterial species was identified to the species level at Day0 in 100% (18/18) of the polymicrobial samples in culture. We also identified two out of two species isolated in culture in 3/18 (17%) of the polymicrobial specimens. This last point is of major importance for the early adaptation of the antibiotic regimen and for prognosis of patients infected with at least two bacterial species. Our protocol might be not fully reliable for polymicrobial infections as it misses at least one type of bacteria in the majority of polymicrobial samples. In our study, exclusive of *S. mitis* group identification that is known to be difficult by MALDI-TOF MS as shown in other studies [9,10,11,12], we did not report any misidentification at Day0. The various sources of identification failure at Day0 were either a complete loss of mass signal, a low log score value that did not allow an identification to the species level, or mixed culture (log score < 1.5). These failures could have been engendered by the high number of leukocytes in some hemorrhagic or purulent/viscous ascitic fluids as previously shown [21]. One explanation is that bacteria are trapped in the gel-like mass of DNA released by leukocytes during the initial lysis step and our 5 min extraction protocol might not be sufficient to overcome this difficulty [21].

The before-and-after comparative study in two populations of 41 patients was performed to assess the real impact of our direct MALDI-TOF identification method on the decrease in time to first antibiotic change in clinical practice for the management of patients diagnosed with SBP. This analysis showed that the time to first change between antibiotics was significantly shortened (decrease of 17 h, *p* < 0.0001 (****)) after the implementation of the method and the percentage of antibiotic adaptation at Day0 increased significantly (increase of 22%, *p* = 0.02(*)) (Table 2). These promising results could have a major impact in clinical practice for the management of the patients. Indeed, SBP is a very common bacterial infection in patients with cirrhosis and ascites [1,2,3,4]. When first described, its mortality exceeded 90% but is has been reduced to approximately 20% with early diagnosis and treatment [6,22]. Therefore, the patient diagnosed with SBP received an empirical treatment as soon as the diagnosis is settled to improve patients’ outcomes. According to European expert recommendations, this antibiotic treatment consists of intravenous third generation cephalosporins or amoxicillin and clavulanic for community-acquired infections, and piperacillin associated with tazobactam for nosocomial infections [5,7]. The American Association for the Study of Liver Diseases, recommend third generation cephalosporins for community-acquired infections and for nosocomial infections: i) piperacillin/tazobactam associated with vancomycin (or daptomycin if known VRE in past or evidence of gastro-intestinal colonization), or ii) meropenem with vancomycin or teicoplanin (if known to harbor MDR gram-negative organisms) [23]. Empirical antibiotic treatments are established on the basis of microbial reports of bacterial cultures and antibiograms that determine bacterial sensitivity patterns. In some specific septic situations, such as pyogenic liver abscess, empirical treatment can be particularly effective because patients treated empirically have similar outcomes compared with patients treated with targeted antibiotics [24]. However, the empirical antibiotic regimens based on epidemiological reports cannot be, in each clinical case, always adapted to the bacteria responsible for the SBP. As previous studies have shown that an early and appropriate therapy active against the isolated bacteria could improve patient outcome, our fast and reliable identification method is of major interest [7,18]. Among the 41 patients diagnosed with SBP after implementation of our protocol, 37% (15/41) benefited from an early adaptation of antibiotic regimen and mean time to first antibiotic was 24.3 h. Interestingly, for 60% of these patients (9/15) the change consisted of an adaptation of the empirical antibiotic therapy active against the identified bacteria. As shown in Table 3, in several cases, direct MALDI-TOF identification had an incremental value compared with Gram staining alone on BCBs for early antibiotic adaptation. For example, patient 9 received imipenem as a probabilistic treatment for nosocomial SBP complicating a biliary cystadenoma. MALDI-TOF at Day0 successfully identified *E. faecium* along with *E. coli,* both to the species level, which allowed the early addition of daptomycin active against the *E. faecium* isolated in culture. Patient 8 had SBP on ethylic liver cirrhosis treated by cefotaxime. Identification of *Bacteroides fragilis* at Day0 allowed a reduction in time to result of 48 h and the early addition of metronidazole active against anaerobic bacteria. For 6 out of 15 patients (40%) the change in antibiotic regimen consisted of a rapid de-escalation that could prevent gut microbiome disturbance and secondary emergence of multi-drug resistance in the digestive tract. This is a crucial point, as late recurrences of SBP caused by resistant bacteria in cirrhotic patients worsen the patients’ prognosis and increases mortality [25]. Finally, in our series, most of the patients (67%) had a favourable clinical outcome at one month following early adaptation of antibiotic therapy. However, five patients had complications in spite of appropriate adaptation of the antibiotic course. This last point could be explained by the numerous comorbidities of these patients.

## 5. Conclusions

In conclusion, our direct identification protocol for positive BCBs inoculated with ascitic fluids is fast (10 min), reliable for monomicrobial infections and inexpensive (EUR 0.4 per test). Its routine integration into a microbiology laboratory allows the early introduction of appropriate antibiotic therapy and can improve the management of patients with SBP. Our protocol might not be fully reliable for polymicrobial infections as it misses at least one bacteria in the majority of polymicrobial samples. Further studies with a larger sample size, including polymicrobial samples, and a longer follow-up period will be of major interest to reinforce our findings.

## Figures and Tables

**Figure 1 microorganisms-10-01188-f001:**
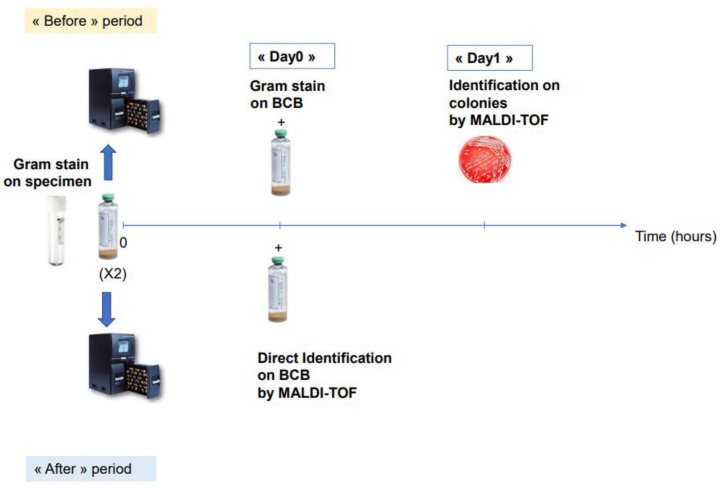
Schematic representation of the comparative study of “before” and “after” direct MALDI-TOF on positive BCBs.

**Table 1 microorganisms-10-01188-t001:** Direct bacterial identifications on ascitic fluid on Day0 by type of bacteria (log(score) ≥ 1.5) *.

Groups	Total No.	No. Concordant	% Concordant
Gram-positive bacteria	98	71	73%
Gram-negative bacteria	58	50	86%
**Total**	**156**	**121**	**78%**
*Staphylococcus aureus*	9	9	100%
Coagulase-negative Staphylococci	37	26	70%
**Total**	**46**	**35**	**76%**
Streptococci	13	5	39%
Enterococci	25	21	84%
Other Gram-positive cocci ^1^	4	2	50%
**Total**	**42**	**28**	**67%**
*Enterobacteriaceae*	44	40	91%
*Pseudomonas aeruginosa*	9	5	56%
*Bacteroides fragilis*	3	3	100%
**Total**	**56**	**48**	**86%**
Aerobic Gram-positive bacilli ^2^	8	7	88%
Anaerobic Gram-positive bacilli ^3^	2	1	50%
**Total**	**10**	**8**	**80%**
*Moraxella osloensis*	2	2	100%
**Total**	**2**	**2**	**100%**

* ^1^ (*Aerococcus urinae*, *Anaerococcus murdochii*, *Rothia mucilaginosa*), ^2^ (*Listeria monocytogenes*, *Corynebacterium* sp., *Bacillus cereus*), ^3^ (*Propionibacterium acnes*, *Clostridium tertium*). * Discordant identifications were caused by direct identification failures. Misidentifications were only reported for S. mitis group species as previously shown [9,10,11,12].

**Table 2 microorganisms-10-01188-t002:** Before-and-after comparative analysis: spontaneous bacterial peritonitis.

	Before Period	After Period	Statistical Analysis (+)
Number of patients	41	41	
Sex ratio (male/female)	33/8	28/13	*p* = 0.2 (ns)
Age	65 +/− 11	62 +/− 11	*p* = 0.32 (ns)
Monomicrobial samples	37/41	37/3	*p* > 0.99 (ns)
Polymicrobial samples	4/41	4/41	*p* > 0.99 (ns)
Mean time for bacterial growth in BCBs	26.4	25.6	*p* = 0.76 (ns)
Enterobacteriacae	17/45	17/45	*p* > 0.99 (ns)
*Enterococcus* sp./*Streptococcus* sp.	18/45	14/45	*p* = 0.37 (ns)
*Staphylococcus* spp.	5/45	7/44	*p* = 0.53 (ns)
Non-fermenting *Bacilli*	3/45	3/45	*p* > 0.99 (ns)
Anaerobic bacteria	1/45	2/45	*p* = 0.55 (ns)
Other bacteria	1/45	2/45	*p* = 0.55 (ns)
Total % of change in antibiotic treatment	15/41 (37%)	15/41 (37%)	*p* = 0.99 (ns)
% change in antibiotic treatment at first result at Day 0 on BCBs (Gram before vs identification after)	6/41 (15%)	15/41 (37%)	*p* = 0.02 (*)
Mean time to first change in antibiotic (hours)	41.3	24.3	*p* < 0.0001 (****)

(+) The unpaired student’s *t*-test was used to compare the quantitative variable, i.e., (age and mean time to first antibiotic change) and the chi-square test was used to compare the categorical variable, i.e., (sex ratio and % of antibiotic change).

**Table 3 microorganisms-10-01188-t003:** Clinical and microbiologic features of patients with SBP.

Patient	Gender	Age	Clinical Features	First Antibiotic	Leukocyte Count (Cells/mm^3^)	GRAM on Positive BCBs (D0)	Identification (Day 0)	Time to First Change in Antibiotic (Hours) (T0)	Antibiotic Optimization	Identification (Day 1)	Outcome
**Patient 1**	Male	58	SBP in a patient with ethylic liver cirrhosis	Tazocillin	11,900	Gram-positive rod	*L. monocytogenes*	20	Amoxicillin	*L. monocytogenes*	Favorable evolution after 10 days of amoxicillin
**Patient 2**	Female	75	SBP in a patient with HCV-related liver cirrhosis	Amoxicillin and clavulanic acid	400	Gram-negative rod	*E. cloacae*	24	Cefepime and metronidazole	*E. cloacae* and *P. aeruginosa*	Not favorable after 5 days of cefepime and metronidazole switched for imipenem.
**Patient 3**	Male	83	SBP	Cefotaxime	315	Gram-positive cocci in chain	*E. faecalis*	18.5	Amoxicillin	*E. faecalis*	Favorable evolution after 10 days of amoxicillin
**Patient 4**	Male	77	SBP	Cefotaxime	5940	Gram-positive cocci in chain	*E. faecalis*	20	Amoxicillin	*E. faecalis*	Favorable evolution after 10 days of amoxicillin
**Patient 5**	Female	67	SBP in a patient with endometrial cancer	Tazocillin	2700	Gram-negative bacilli	*E. cloacae*	18.5	Cefepime	*E. cloacae*	Favorable evolution after 7 days of cefepime
**Patient 6**	Male	67	SBP in a patient with ethylic liver cirrhosis	Tazocillin	270	Gram-negative bacilli	*E. coli*	24	Cefotaxime	*E. coli*	Favorable evolution after 4 days of cefotaxime followed by a oral amoxicillin and clavulanic acid for a total of 7 days
**Patient 7**	Female	85	SBP in a patient with secondary liver involvement by lymphoma	No antibiotic	250	Gram-positive cocci in clusters	*S. aureus*	22	Introduction of cefazolin	*S. aureus*	Not favorable. Death 10 days after antibiotic initiation.
**Patient 8**	Male	54	SBP in a patient with ethylic liver cirrhosis and bleedings of varices. Past medical history of SBP caused by *S. pneumoniae*	Cefotaxime	22,140	Gram-negative bacilli	*B. fragilis*	69	Addition of metronidazole	*B. fragilis*	Not favorable. Recurrence of SBP at 5 days of antibiotic initiation and switched for imipenem
**Patient 9**	Male	71	SBP and sepsis in a patient with ethylic liver cirrhosis.	Imipenem	1200	Gram-positive cocci in chains and Gram-negative bacilli	*E. coli* and *E. faecium*	21.5	Addition of Daptomycin	*E. coli* and *E. faecium*	Not favorable. Patient died 7 days after appropriate antibiotic treatment
**Patient 10**	Male	67	SBP in a patient hospitalized for drainage of refractory ascites caused by *K. pneumoniae* producing ESBL.	No antibiotic	315	Gram-negative bacilli	*K. pneumoniae*	16	Initiation of imipenem	*K. pneumoniae*	Not favorable. Death 19 days after antibiotic initiation.
**Patient 11**	Male	52	SBP in a patient hospitalized for hepatocellular carcinoma	Tazocillin	250	Gram-positive bacilli	*L. monocytogenes*	17.5	Amoxicillin	*L. monocytogenes*	Favorable after 7 days of amoxicillin
**Patient 12**	Male	60	SBP in a patient with peritoneal carcinomatosis and treated by cefoxitine for a PICC line infection	Cefoxitine	110	Gram-positive cocci in chain	*E. faecalis*	17	Amoxicillin	*E. faecalis*	Favorable after 7 days of amoxicillin
**Patient 13**	Male	57	SBP in a patient with acute liver failure complicating a primary sclerosing cholangitis	No antibiotic	210	Gram-positive cocci in chain	*S. anginosus*	20	Amoxicillin	*S. anginosus*	Favorable after 7 days of amoxicillin
**Patient 14**	Male	54	SBP in a patient with ethylic liver cirrhosis	No antibiotic	550	Gram-positive cocci in chain	*S. gallolyticus*	39	Tazocillin	*S. gallolyticus*	Favorable after 24 h of tazocillin followed by a total of 10 days of oral amoxicillin
**Patient 15**	Male	60	SBP in a patient with ethylic liver cirrhosis	No antibiotic	300	Gram-positive cocci in clusters	*S. aureus*	17	Amoxicillin and clavulanic acid	*S. aureus*	Favorable after 10 days of amoxicillin and clavulanic acid

## Data Availability

Data are available upon reasonable request.

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
