# Peer review of "Spontaneous Bacterial Peritonitis: The Incremental Value of a Fast and Direct Bacterial Identification from Ascitic Fluids Inoculated in Blood Culture Bottles by MALDI-TOF MS for a Better Management of Patients"

_microorganisms, 2022, doi:10.3390/microorganisms10061188_

Round 1

Reviewer 1 Report

Lotte and co-authors present an interesting and important study on more rapid identification of bacteria causing peritonitis by direct MALDI-TOF analysis. As the authors write, improvements in bacterial identification are needed to tailor antibiotics, increase patient survival and decrease antibiotic resistance development.

While data on direct identification of bacteria are convincing, I fail to understand the before- after part of the study. E. g. I do not understand, how exactly 39 patients were selected, why there is a time gap of several years between before and after and why samples with invalid results in the after period were excluded from analysis. All these factors may favour bias towards a more positive result in the after period. I suggest that the authors better describe the process, e. g. through a flow-chart, or include all available samples in the comparison and not only 39.

Also, it seems not appropriate to state that lack of inappropriate narrowing of antibiotic treatment was due to physicians not reacting to the test results or to the fact that it would have occurred in bacterascites not needing treatment.  A new test must stand by its own; you can not guarantee that more or less by chance or external factors misinterpretation of the results will not occurr.

Minor issues are that it may be better to talk about bacterial peritonitis and not spontaneous bacterial peritonitis, because this is a term for very specific primary forms of peritonitis which does not seem appropriate e.g. for patients 5 and 12. In addition, caption of table 3 is not a correct phrase. Finally, the recommendations by AASLD are not reported correctly in the discussion.

Author Response

Lotte and co-authors present an interesting and important study on more rapid identification of bacteria causing peritonitis by direct MALDI-TOF analysis. As the authors write, improvements in bacterial identification are needed to tailor antibiotics, increase patient survival and decrease antibiotic resistance development.

We thank the reviewer for this comment.

While data on direct identification of bacteria are convincing, I fail to understand the before- after part of the study. E. g. I do not understand, how exactly 39 patients were selected, why there is a time gap of several years between before and after and why samples with invalid results in the after period were excluded from analysis. All these factors may favor bias towards a more positive result in the after period. I suggest that the authors better describe the process, e. g. through a flow-chart, or include all available samples in the comparison and not only 39.

We thank the reviewer for these comments. The initial description of the method of the before en after case control study was may be not adequately and sufficiently described, and we clarified the methodology in the revised version of the manuscript.

-First, regarding the number of patients included in the before and after case control study. As the aim of the study was to evaluate the reduction in antibiotic optimization turnaround time allowed by the implementation of our direct identification method, we only included the 41 infected patients (among 61 patients) that were diagnosed with a SBP and therefore treated by antibiotics. The 20 patients diagnosed with bacterascites were logically not included as they did not receive antimicrobial treatment, so for them no reduction in antibiotic optimization TAT cannot be evaluate. More, we have withdrawn 2 infected patients (out of 41) for whom direct identification failed at Day0 in the initial version of the study. As the reviewer suggest that this exclusion may introduce bias in the analysis, we added again these two patients in the revised version of the analysis performed on 41 patients.

To explain these points we have made modifications in the revised version of the manuscript:

Methods section 2.4: the following modifications have been made

° The following section was modified in the revised version of the manuscript: “To evaluate….in the comparison”(new lines 118/122)

° The following sentence has been withdrawn of the first submitted version of the paper:  ”Patients for whom direct…decision at Day0” (lines 119-122 in the first submitted version)                                                                                                                                                                                                                              

Result section 3.2: the following modifications have been made

° The following sentence was added in the revised version of the manuscript: “The twenty patients….antibiotic treatment (new lines 181/182)

° The following sentence has been withdrawn of the first submitted version of the paper:  ” Among the 41 patients, 2/41 (5%) were excluded because MALDI-TOF identification failed at Day0. The remaining 39 patients (95%) diagnosed with SBP were included in the before-and-after case control study (lines 172/174 in the first submitted version)

In the new version, we included all the 41 patients with SBP treated by antibiotic in the comparative analysis, and this do not change the significance of our results (see modifications in new table 3).

-Second, concerning the time gap of two year in our before and after comparative study. The direct identification method on BCBs was implemented in our laboratory in 2017. In the first two years (2017-2018), the method was being validated and not systematically performed on all ascitic fluid samples incubated in BCBs. When our study on Direct MALDI-TOF identification on blood culture sample was published in January 2019, we start to perform Direct MALDI-TOF on all ascitic samples incubated in BCB.        

Therefore in our comparative analysis that aim to evaluate the impact of our method in reduction in antibiotic optimization                                        1) the before period was a two year period 2015-2016 when direct identification was never performed on ascitic fluid inoculated in BCBs     2) the after period was a two year period 2019-2020 when direct identification was systematically performed on all ascitic fluids inoculated in BCBs            

The 41 infected patients included in the after period were compared to 41 infected patients treated by antibiotic in the before period included in a chronological order.                                                                                                                                          

Also, it seems not appropriate to state that lack of inappropriate narrowing of antibiotic treatment was due to physicians not reacting to the test results or to the fact that it would have occurred in bacterascites not needing treatment.  A new test must stand by its own; you cannot guarantee that more or less by chance or external factors misinterpretation of the results will not occur.

We thank the reviewer for this comment concerning our explanation of the lack of inappropriate narrowing of antibiotic treatment for polymicrobial samples in our study. If this explanation was right for some patients in the study, we assume that it may be a bit of overstatement for patient in general. Another study including a larger number of polymicrobial samples could be performed to evaluate the impact of our direct identification method on potential inappropriate narrowing of initial empirical wide spectrum antibiotic treatment.

Therefore, we removed the following sentences in the discussion and conclusion section from the initial version of the manuscript “However in this study…antibiotic treatment” (lines 227-231 discussion and lines 270-273 conclusion in first submitted version). We have stated in the revised discussion and conclusion sections that polymicrobial samples constitute a potential limit of our methods and another study including a larger number of polymicrobial sample can be performed (new lines 463-464)

Minor issues are that it may be better to talk about bacterial peritonitis and not spontaneous bacterial peritonitis, because this is a term for very specific primary forms of peritonitis which does not seem appropriate e.g. for patients 5 and 12.

After an in-depth review of patient 5 and 12 medical files we confirm that these patients were diagnosed with Spontaneous bacterial peritonitis and not secondary peritonitis.

Our article is focused on diagnosis and treatment of SBP.

In addition, caption of table 3 is not a correct phrase.

We have modified the caption of the new table 3

Finally, the recommendations by AASLD are not reported correctly in the discussion.

Below you will find the recommendations of the AASLD 2021 as I have found in table 9 :Biggins et al. . Diagnosis, Evaluation, and Management of Ascites, Spontaneous Bacterial Peritonitis and Hepatorenal Syndrome: 2021 Practice Guidance by the American Association for the Study of Liver Diseases. Hepatology 2021, 74, 1014–1048, doi:10.1002/hep.31884.

To report correctly these recommendations, we have modified the following sentence in the discussion section (new lines 435-439).

Reviewer 2 Report

I read with interest the paper that describes using ascitic fluid samples from spontaneous bacterial peritonitis patients with intent for early microbial detection for targeted antimicrobial therapy that could potentially improve clinical outcomes. Authors show correct identification of some bacteria and describe how in patients 8/9 they could start targeted antibiotic therapy early. Authors also acknowledged challenges in polymicrobial sepsis patients but argue that this may not be clinically very impactful. Authors mention cost data and claim very fast results that impact clinical care. 

Intuitively it makes sense to sample ascitic fluid and it is likely that yield will be higher than blood culture. I have two comments to authors:

1. Every hospital has antibiogram reported on annual basis. This states which bacteria is isolated from which fluid and its sensitivity pattern. This information is essential for "empirical" antibiotic therapy in septic patients like spontaneous bacterial peritonitis patients. It is also shown that patients who are treated empirically have similar clinical outcomes as compared to patients who are treated with targeted antibiotics - PMID: 27733320. 

So please add this information and discussion in your manuscript. 

2. You claim that you could diagnose early and in many patients you had to change initial antibiotic regimen (for good reasons). If your initial antibiotic choice was based on empirical evidence of bacterial isolates, you would not have to change antibiotic therapy for so many patients. Can pls explain this results. 

3. Changing antibiotics and saving time of x hrs; does it translate into reduction of 30-day or in-hospital mortality? 

Author Response

I read with interest the paper that describes using ascitic fluid samples from spontaneous bacterial peritonitis patients with intent for early microbial detection for targeted antimicrobial therapy that could potentially improve clinical outcomes. Authors show correct identification of some bacteria and describe how in patients 8/9 they could start targeted antibiotic therapy early. Authors also acknowledged challenges in polymicrobial sepsis patients but argue that this may not be clinically very impactful. Authors mention cost data and claim very fast results that impact clinical care. 

We thank the reviewer for these comments.

Intuitively it makes sense to sample ascitic fluid and it is likely that yield will be higher than blood culture. I have two comments to authors:

  1. Every hospital has antibiogram reported on annual basis. This states which bacteria is isolated from which fluid and its sensitivity pattern. This information is essential for "empirical" antibiotic therapy in septic patients like spontaneous bacterial peritonitis patients. It is also shown that patients who are treated empirically have similar clinical outcomes as compared to patients who are treated with targeted antibiotics - PMID: 27733320. 

So please add this information and discussion in your manuscript. 

  1. You claim that you could diagnose early and in many patients you had to change initial antibiotic regimen (for good reasons). If your initial antibiotic choice was based on empirical evidence of bacterial isolates, you would not have to change antibiotic therapy for so many patients. Can pls explain this results. 

Comments 1 and 2 are interesting. It is true that protocol of empirical antibiotic based on antibiogram reported (for example annual reports) in tertiary care centers along with empirical antibiotic protocol based on international guidelines can help to define an appropriate antibiotic treatment.

To to take into account these two comments, and to add the information that patients with PLA treated empirically have similar clinical outcomes as compared to patients who are treated with targeted antibiotics (Shelat et al., we have added the following sentences in the revised version of the manuscript: ”Empirical antibiotic …antibiotics “ (new lines 439-442)

However, the empirical antibiotic regimens based on epidemiological reports cannot be, in each clinical cases in real life, always adapted to the bacteria responsible for SBP. For some reasons such as high proportion of migrant in some territories (such as in our tertiary care center in Nice), drug-resistant bacteria can emerge quickly in hospital populations and empirical antibiotic therapy might not be active on the strain isolated in culture in each clinical cases. Therefore the empirical initial antibiotic might be switched in such cases. In some other cases in the study, the changes consisted in rapid de-escalation that can prevent gut microbiome disturbance and secondary emergence of multi-drug resistance in the digestive tract

  1. Changing antibiotics and saving time of x hrs; does it translate into reduction of 30-day or in-hospital mortality? 

This is an interesting question. However, our study was not designed to answer to this question. This is a limit of our study .Large-scale epidemiological studies will be of major interest to answer to this question.

Round 2

Reviewer 1 Report

I am grateful that the authors answered all my concerns extensively and in depth. There is only one issue that must be resolved before the manuscript may be published: the AASLD guidelines on treatment recommendations for SBP are not cited correctly. It is true that the authors cite the table correctly, but the table contains a spelling error, which becomes obvious by logical reasoning and from reading the text (page 1030 at the top): either Pip/Taz plus minus vancomycin (daptomycin in case of previous VRE) or meropenem plus minus vancomycin / teicoplanin.  The combination of pip/taz plus meropenem does not make sense; in the table, "and" and "Or" have to be switched.

Author Response

I am grateful that the authors answered all my concerns extensively and in depth.

We thank the reviewer for this comment.

There is only one issue that must be resolved before the manuscript may be published: the AASLD guidelines on treatment recommendations for SBP are not cited correctly. It is true that the authors cite the table correctly, but the table contains a spelling error, which becomes obvious by logical reasoning and from reading the text (page 1030 at the top): either Pip/Taz plus minus vancomycin (daptomycin in case of previous VRE) or meropenem plus minus vancomycin / teicoplanin.  The combination of pip/taz plus meropenem does not make sense; in the table, "and" and "Or" have to be switched.

We thank this error notification. Indeed the combination of piperacillin/tazobactam and meropenem does not make sense.
We have made the modification in the revised version of the manuscript (new lines 244-247).